# European Integration Processes in the EU GI System—A Long-Term Review of EU Regulation for GIs

**Karola Schober** [1,2,*], **Richard Balling** [2], **Tobias Chilla** [1] and **Hannah Lindermayer** [2]

1    Institute of Geography, University of Erlangen-Nuremberg, 91058 Erlangen, Germany
2    Bavarian State Ministry for Food, Agriculture and Forestry, 80539 Munich, Germany
*    Correspondence: karola.schober@stmelf.bayern.de

**Abstract:** *Prosciutto di Parma*, *Bavarian beer* and *Roquefort*—Geographical Indications (GIs) have been systematically protected at the EU level for 30 years and are now an important part of the *farm2fork strategy*. The article analyses how the integration of the EU GI system can be explained from an institution and discourse theoretical perspective and ties in with the soft spaces debate. In doing so, scalar shifts in competence from a German perspective and the role of discursive spatial relations are examined in more detail. The empirical results are based on a mix of methods that includes the evaluation of secondary statistical data, document analyses, participant observation and expert interviews. The study shows that the European Commission (EC) is increasingly acting as a spokesperson for GIs, but that regional actors are also playing a more important role in implementation and enforcement. This development is fed by the influence of the agricultural policy instrument in terms of competition, but also consumer protection and trade policy. Overall, there are three development layers: protect and systematise, legitimise and expand and open and defend. A more independent development of the EU GI system as an instrument of quality policy and for the development of rural areas could give greater weight to the sustainability-relevant, environmental policy aspects currently demanded by society.

**Keywords:** Geographical Indications; regional products; regional food; CAP; European integration; institutional spillover; soft spaces

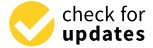

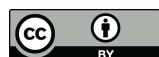

## 1. Introduction: Origin Labelling of Food in Europe

Geographical Indications (GIs) have been systematically protected at the EU level for 30 years and are now an important part of the farm2fork strategy. The three decades of policy formulation and implementation have to be understood as an incremental process. A high number of actors, institutions and interests have been interacting across the EU member states and throughout the changing periods of political priorities.

Our paper analyses this process and intends to explain the development layers. This obviously involves questions of mandate and power [1–6]: who is allowed to determine the "rules of the game"? What kind of resources are made available? What can shifting competences to one level mean for others? These questions are of particular complexity as the involved EU member states are characterised by their different food traditions and political cultures. The EU multi-level governance system is a challenging policy area [7–9], and in the GI case, it involves a series of sectoral policies such as the Common Agricultural Policy (CAP), trade and economic policy (in particular the European Single Market (ESM). We approach the complexity of these longstanding processes by means of a mixed methods approach, based on the Reflexive Grounded Theory [10].

Moreover, the protection regime as such comes along with complex spatial implications. For each product, the concrete perimeter has to be defined and also the specific ingredients or production methods that are bound to these areas. We understand the involved

processes of geographical regulations as regionalisation processes [11]. This means the political negations processes of scales and perimeters come along with a certain "hardening" of spaces and their roles. Our analysis aims to explain these processes of geographical definitions, mainly based on the "soft spaces" approach [1–3,6,12].

The article reconstructs and analyses interdependencies of effects, asks about the use of discursive spatial references and sees European integration processes as being highly shapeable by regional and national actors. The article analyses the extent to which regional and national influences affect (de)stabilisation processes at the European level. In doing so, this contribution closes a gap in Europeanisation studies, especially with respect to GIs.

The structure of this paper is as follows: Section 2 summarises the state of academic debate regarding European Policy and national practices on origin labelling in a broader sense and the present understanding of processes of European integration. Section 3 clarifies the research methodology of this paper. Thereafter, in Section 4, the results are reported. A discussion regarding the research findings, its limits and future avenues is given in Section 5. Finally, Section 6 concludes this study by showing the limits of the work and a brief outlook pointing to the potential for continued independent development of the EU GI system.

## 2. Literature Review

The EU GI system is addressed by various scientific disciplines even if the agricultural economic perspective is of particular prominence here. This is also because the EU GI system goes back to French wine law and involved the idea of the terroir [13–17].

Not only in France but also in other European, mainly Romance countries, national protection systems with a focus on origin developed [16]. Increasing world trade questioned the efficiency of national protection systems, as many imitations appeared outside the national territory–which is why a European protection system was created in 1992 [18] (p. 1).

In Germany, GI research started in the early 2000s from a mainly agricultural economic view [19–22].

Today the economic benefit analysis of GI protection is more interdisciplinary [23,24], and also more strongly related to its collective character [21,25], trade with third countries [26–29] and its innovation potential [30]. Additionally, the GI system is discussed scientifically in the context of rural development, location development or spatial and regional development, or in terms of regional value chains [25,31–33], mostly case study-based [34–36]. The (agricultural) economic perspective mainly examined consumers' willingness to pay, price premium, market size and rural development (according to a meta-study [37]). More generally, it can be stated that information on origin leads to a purchase or additional price argument for consumers. This has been confirmed many times by country-of-origin research (first [38], more recently, e.g., [39]). With smaller-scale references, price effects in the area of origin protection have also been proven on the basis of case studies [37,40–42]. In recent years the field of social and cultural science has focused on changeability and its impact on governance [43–49]. Recently, Belletti and Marescotti (2021) have published guidelines for context-sensitive evaluation of GIs [50].

The discourse on GIs is accompanied by a legal debate [17,51–53] that reveals questions of demarcation and discusses the edges of the legal framework as well as illustrating the changeability of *hard* institutions.

At the European level and in the EU member states, various options for origin labelling have developed. Conceptually, these are distinguished between *simple*, *combined* and *qualified* types of origin-related designations (see Table 1) [14].

**Table 1.** Types of origin-related designations and their association with quality.

| Types of Origin-Related Designations | Simple | Combined | Qualified |
|---|---|---|---|
| The Link between Origin and Quality |  |  |  |
| Examples | "Made in France", "Pork and beef completely from Hesse", "Packed in 80801 Munich", German "Regionalfenster" | Label Rouge (France), Bayerisches Bio-Siegel (Germany), Red Tractor (UK) | Prosciutto di Parma PDO (Italy), Bavarian Beer PGI (Germany) |

Source: [14] p. 2ff, adapted and supplemented.

As Table 1 shows, in the case of simple origin-related designations, the message of origin stands on its own without a quality statement, but it can be ethnocentrically charged and thus interpreted as *better because of origin*. If the quality is functionally justified, it is a *qualified* indication of origin, e.g., in the sense of the EU GI system. In the case of combined origin-related designations, by definition, there does not have to be a functional link between quality and origin (typicity). In the case of combined origin-related designations, the quality is rather to be seen as an additional purchase argument that stands next to the message of origin [14]. For reasons of state aid law, the quality must be in the foreground; the origin is secondary and in principle can be combined flexibly.

Origin-related designations can also be categorised according to whether they are obligatory or optional from the producer's point of view and whether they were initiated by the state or privately (Table 2).

**Table 2.** Examples of origin labelling: classification proposal.

| | Obligatory | Optional |
|---|---|---|
| Public Law | Special regulations for meat according to Regulation (EU) No 1169/2011, national regulations for certain products (e.g., for milk in Italy and France) | GIs according to Regulation (EU) No 1151/2012, EU-notified national quality (and origin labelling) systems |
| Private Law | none | Regionalfenster (in Germany) |

Source: own representation.

In principle, origin labelling is voluntary for companies in the EU. However, the regulation regarding food information for consumers (EU) No 1169/2011 contains a whole series of obligatory special regulations, the main aim of which is to guarantee food safety (e.g., in the sensitive beef sector; keyword: BSE) [54]. In addition, within the framework of the marketing standards (Regulation (EU) No 589/2008 [55]), the indication of origin by means of a code is obligatory for eggs (e.g., 1-DE-1234567 = free-range farming method, country of origin Germany, identity of the laying farm with house number). The name, company name or address of the producer is not considered an indication of origin but is mandatory (Regulation (EU) No 1169/2011 Art. 2(2)(g)). These obligatory regulations are driven by consumer policy; they are therefore intended to promote traceability and thus food safety. The *farm2fork strategy* was also born out of a similar idea; approaches can already be found in the beginnings of the so-called *General Food Law* (Regulation (EC) No 178/2002 [56]).

Regardless of whether the indication of origin is voluntary or obligatory, the following applies: the consumer must not be misled (Regulation (EU) No 1169/2011 Art. 7(1)(a)). As soon as the product suggests a certain origin (and this impression could also be created by the use of symbols typical for the country such as the national flag, the national colours

and famous buildings), the actual origin must be indicated, so to speak, as a clarifying or de-localising indication (Art. 26(2)(a)).

In contrast, (national) state activities aimed at highlighting a specific origin are in principle critically discussed at the European level due to their internal market-distorting character (TFEU Art. 28, Art. 30, Art. 107(1)). France was able to introduce national mandatory origin labelling for milk in 2017 as part of a two-year test phase, which was extended unilaterally, followed by seven other countries (including Italy, Portugal and Spain) [57,58]. However, the European Court of Justice (ECJ) clarified on 1 October 2020 (Case C-485/18) that when introducing mandatory origin labelling, there must also be a demonstrable link between quality and origin; in France's specific case, the national origin labelling requirement was therefore anti-competitive. Nevertheless, France has enforced mandatory origin labelling for certain meat dishes in community catering in 2022 and Sweden has also launched a similar initiative [59,60]. This also indicates that the discussion on whether origin labelling should be regulated nationally or at the European level has recently gained momentum again.

State programmes related to origin fall into the area of voluntary measures. They also provide the opportunity for state support for certification, controls and consumer communication. However, as state aid for certain locally bound producers, these are considered to have a market-distorting effect under European law (TFEU Art. 28, Art. 30, Art. 107(1)). These state programmes are only accepted by the EC if there is an explanation under state aid law for the corresponding market intervention. The limits are currently set by the Guidelines for State Aid in the Agriculture and Forestry Sector and in Rural Areas 2014–2020 (2014/C 204/01) and the Agricultural Block Exemption Regulation (Regulation (EU) No 702/2014 [61]).

In addition, the origin may only be secondary to the quality message. This means that, in EU logic, state programmes of origin are considered, rather, as quality programmes, i.e., quality marks or seals with a subordinate message of origin in the labelling sense. They are supported or promoted by the state and are offered to companies as voluntary labelling options (e.g., Label Rouge (France), AMA-Qualitätszeichen (Austria), Bayerisches Bio-Siegel (Germany)). In Germany, in addition to other regional quality programmes, there is the Regionalfenster (regional window), a state-initiated but privately funded system of origin labelling. This provides a standardised space to indicate optional indications of origin (without reference to quality)—a voluntary declaration field [62].

In contrast to all these options of state (quality and) origin labelling, which are generally viewed rather critically by European legislation, the promotion of GIs (PGI/PDO) is recognised and worthy of support at the European level: European resources, primarily from the agricultural budget, but also from the Structural Funds, flow into financial instruments (such as certification, sales and regional promotion). This tension between competitive and primarily agricultural policy goals raises the question of integration processes in the GI system.

This article shows how the GI system with its functional terroir logic, originally a French-influenced policy upload, was integrated at the EU level and has since been further developed and strengthened vis-à-vis other national systems as well as international trademark law [63] (p. 151ff).

The EU GI system is part of overarching Europeanisation and European integration processes [64]. Basically, Europeanisation is seen as the successful adoption of European values at the sub-European level, which is expressed above all in the implementation of European regulations and programmes in the member states [8] (p. 38ff). Europeanisation is thus the consequence of European integration at the sub-European level [8] (p. 38). European integration is the strengthening of institutional units at the European level [8] (p. 38ff)—in principle independent of their democratic legitimacy. A large number of authors deal with integration dynamics (e.g., [65–69], but the opposing concept of disintegration has also been studied (e.g., [70,71]).

These interwoven processes of European integration and Europeanisation can also be seen as a necessary product (and perhaps a historical intermediate outcome) of globalisation [72] (p. 9). The underlying assumption is that they happen, or better, are made, procedurally and revolve around the allocation of power, competences and resources [73] (p. 1).

For the analytical reconstruction of vertical processes of change, the article draws on institutional and discourse theory considerations. This is based on a broad understanding of institutions: i.e., in the development of European GI institutions, not only the organisational units (such as Directorates-General and associations) or the regulatory framework were considered, but also the practices of actors (e.g., activities in committees).

The institutional economic game metaphor (according to [74]) serves to describe and explain these processes. In an ideal-typical view, formal institutions can be seen as the rules of the game (e.g., the EU GI regulation), while the game moves as actions (e.g., issue-related cooperation) are to be assigned to informal institutions [75] (p. vii). In practice, the processes of rulemaking and rule application are interwoven. In the legal literature, the basic idea of the processual shaping of structures is also found, especially clearly in Georg Jellinek's (1900) idea of the "normative power of the factual" [76]. There, the changeability of rules on the basis of social reality is also emphasised and the enforcement mechanisms are illuminated (described in more detail in the anthology by [77]).

The sociological institutional theory further suggests focusing on four allocation problems: "competences", "resources", "legitimacy" and "control" [5] (p. 61ff). This model, originating from the 1990s, is accentuated for the present analysis of the EU GI system in the direction of process orientation and emphasis on interaction possibilities in order to emphasise relationalities more strongly. "Interaction" is cited as an additional allocation category because the ability to develop and use networks is seen in innovation sociology as a key element for institutional change processes [78,79].

The basic assumption of this multidimensional analysis model is that institutions developed in a particular space have an effect on that space's organisational stability. Stabilisation processes can even lead to the sphere of influence of this spatial unit expanding beyond its administrative boundaries in individual dimensions (e.g., enforcement of European standards within the framework of free trade agreements).

In this ideal typical view, the legitimisation of these allocation processes by local people is seen as the foundation for these same processes in the *Europeanisation game*, as institutions are seen as the product of social negotiation processes [5] (p. 62).

"Discourse coalitions" [80] (p. 217) are characterised by the fact that a group of actors uses a particular set of narrative strands during a certain period of time. However, the approach does not assume that the supporting actors of discourse coalitions necessarily have to agree to do so [80] (p. 12) or act intentionally (cf. [81] (p. 14)), but rather that it is sufficient that a common work of meaning is carried out in the public debate and shared narrative patterns are claimed [80] (p. 12).

The concept of path dependencies (understood as developmental important pre-structuring, [82]) is suitable for examining significant institutional changes over time. In the analysis of GI processes, it serves to explore the context of action more closely, while avoiding the danger of limiting oneself to trivial "history matters" [82]. The concept of path dependency, first used by the economic mathematician William Brian ARTHUR (1989), has also been used in recent decades to explain processes of social transformation [83,84]—a debate in which the EU GI system has been discussed more recently [85,86].

In this work, the institutional theory approaches find a connection to the geographical debate through the "soft spaces" approach [1–3,6,12]. The idea of examining spatial changes through institutional distribution mechanisms is based, for example, on rescaling studies from spatial and environmental planning [9,87].

Following the (post-)structuralist understanding of space or scale, such vertical processes of change are also socially constructed and changeable over time, become relevant only through their relations and are socially and politically controversial [88].

Soft spaces thus exist along and alongside territorially ordered spaces and scales ("hard spaces" [6]) and are to be understood as a "shorthand for the wide range of governance bodies and strategy-making processes and implementation practices" [6] (p. xxi). The concept is deliberately relatively broad in order to maintain the explorative potential of this "increasingly important research area" [6]. The increasing importance is also due to the fact that throughout Europe "new, non-statutory or informal spaces [ . . . ] can be found in a variety of circumstances and with diverse aims and rationales" [6].

Allmendinger et al., however, see these new spaces as legitimised less through elected representatives and more through a heterogeneous group of political, social and market actors [6] (p. 4). Cohen and McCarthy (2015) also emphasise that decisions are often shifted to a level of scale that is only indirectly legitimised democratically and even see such rescaling processes as an expression of neoliberal restructuring [89].

In their study of the Baltic Sea macroregion, Metzger and Schmitt (2012) identify as an important developmental step the fact that the EC was able to act "as a designated spokesperson for the interests of the wider Baltic Sea Region" [4] (p. 265). They see the establishment of a spokesperson role as an indication of the stabilisation of a spatial unit. They also emphasise the high importance of intersubjective recognition of this quasi-imagined order (similar to [90] (p. 58)).

Similarly, the fading-in and fading-out of spatial categories in political discourse can play a decisive role in the political process [6]. Thus, political processes can get by without spatial references in some phases ("spatial tabooing" [6] (p. 2713). Often, it is precisely this fading out of spatial references at the European level that is a discursive enforcement strategy at a certain point in time. At a later point in time, spatial references become the subject of discussion again, especially when spatial conflicts of use arise ("reterritorialization" [6]).

## 3. Data and Methods

The data originating from the underlying research project were collected and analysed mainly qualitatively and by applying the "Reflexive Grounded Theory" [10]. As is typical for such a research design, the research process was not considered in a linear way, but conceptual considerations and findings from the empirical data collection were put into a reciprocal relationship.

This recursive approach, which actively includes the researcher's point of view, is a hermeneutic principle and should ensure quality [10]. The inclusion of the researcher's point of view also results in the partial focus on the German perspective, e.g., when it comes to explaining scalar shifts. Different methods were thus combined with the aim of complementary compensation [91,92]. The resulting mix of methods is as follows:

1.  Analysis of secondary statistical data, primarily eAmbrosia [93].
2.  Broad document analysis (according to Flick 2012 [92]), especially legal bases and legal evaluations.
3.  Participant observation (according to Mayring 2002 [94], 26 events and expert discussions) and research diary (FTB; according to Breuer et al. 2017 [10]). The events took place in the national context of Germany (15 events) or with a stronger European perspective in Brussels (11 events).
4.  A total of 49 guided, anonymised expert interviews (according to Meuser/Nagel 1991 [95]) at European, national and regional levels with an average duration of 70 min. The interviewees were persons who have been responsible for the conception or implementation of national origin labelling systems in a member state or region or who have been involved in the integration process of the EU GI system (lobbyists, GI experts, (former) representatives of the EC). The interview guide used was increasingly fine-tuned in the selection process of the study, but also modified with regard to the respective person to be interviewed and their scalar and thematic location.

The empirical data were collected between January 2019 and October 2020. Over the equivalent of around two months (29 April to 17 May 2019, 21 October to 25 October 2019 and 27 January to 28 February 2020), many expert interviews and participant observations

took place in Brussels. The intensive study of the legal matter took place mainly in the first half of 2019. All available documents that contained primary or secondary collected data were analysed using the qualitative data and text analysis software MaxQDA. Rule-based techniques and procedures were used to find, construct and elaborate analysis steps during the research process [10].

The corpus was considered relatively complete or meaningful if the data collections on which the results were based did not suggest any new thematic phenomena [96] (p. 130). Basically, the aim was to elaborate a specific vocabulary for reconstructing and making transparent the focused field of action "Institutionalisation Processes and Spatialities in the EU GI System". Such an inductive–deductive approach is not new and can be described as "interpretative-structuralist-pragmatic analysis" [97] or "methodological hybrid" [97]. The more recent developments, i.e., those of the last two years, are also discussed against this background.

Overall, the result of this procedure can best be described as an "object-based theory" [94], i.e., as a "middle-range theory" [95], in other words, its explanatory power is limited to the object of study. For details of this study see [63].

## 4. Results: Stabilisation of the EU GI System over Time

The stabilisation processes of the GI system can be divided into three development phases that lead to overlapping "layers", namely (1) protect and systematise, (2) legitimise and expand and (3) open and defend. These phases are shown in Figure 1 next to the increasing number of GIs from EU Member States (by product group).

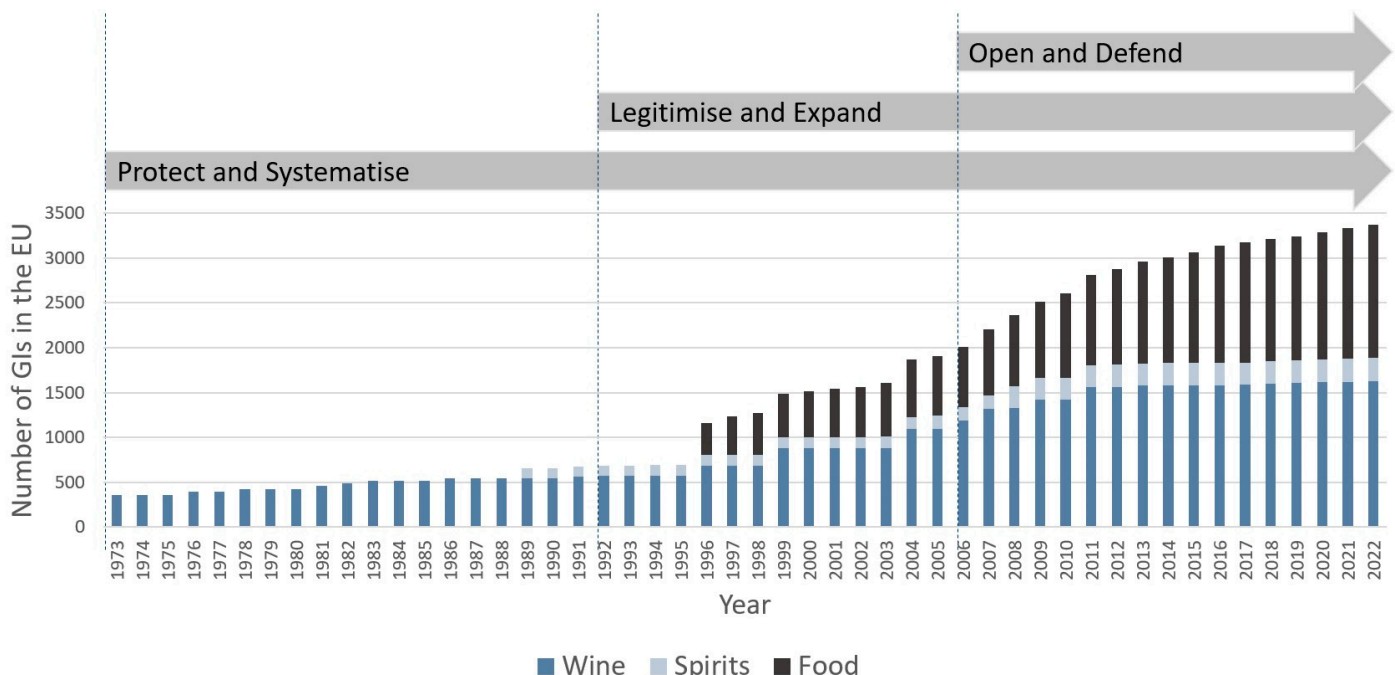

**Figure 1.** Phases in the development of EU GIs into Rural Intellectual Property Rights (IPRs). Source: own presentation. Data source: eAmbrosia (as of 28 December 2022).

Table S1 "Milestones in the development of GI rules for agricultural products and foodstuffs" shows the relevant elements of the development of the GI rules from their introduction (Regulation (EEC) No 2081/1992 [98]) to the current amending regulation under CAP 2023 to 2027 (Regulation (EU) No 2021/2117 [99]) or proposal for the coming reform of the regulation.

### 4.1. Layer 1: Protect and Systematise

The focus on *protection and systematisation* is characterised by the institutional establishment of the possibility of protection for GIs. The adoption of Regulation (EEC) No 2081/1992 (entry into force: 24 July 1993) was the decisive step for a systematised protection. A GI system had already been discussed by the twelve member states of the European Economic Community (EEC) since the mid-1980s within the CAP framework. In 1992, it was then introduced for the first time with its own regulation for the relatively broad field of agricultural products and foodstuffs.

Various path dependencies played a role in the enforcement of the first GI Regulation, such as:

- The traditionally relatively strong regulation of agricultural policy
- Its stronger orientation towards quality instead of quantity since the 1980s
- The European goal of realising a free internal market
- The relatively strong role of the national states in the process ("regulatory committee procedure")
- The not yet developed but in a certain way competing trademark law
- The strengthening ECJ case law
- The existing rules on European level for wines, aromatised wine products and spirits

Especially the historically grown protection rights for wines (since 1970), aromatised wine products (since 1991) and spirits (since 1989) [100] (p. 351) played an important role in the introduction process. These old designation protection rights were originally national protection systems that were now recognised at the European level (expert interviews). Other arguments were made with regard to the increasing consumer demand for quality products as well as the increasing intra-European trade, which required the same standards. These framework conditions laid the foundation for the later expansion of the EC's role as spokesperson for regional products. For more details, see [63] (p. 104ff).

In the 1990s, the GI system was discussed, due to its market intervention character, on the one hand as a contradiction to the declared goal of a free European single market (a rather market liberal argumentation) and on the other hand precisely as a means for the functioning of such a single market (a rather Keynesian argumentation). In other words, the main issue was how to implement the European single market with as uniform standards as possible and how much regulation was needed to achieve this.

The relatively high scope of protection [21] (p. 1031) and the relatively strong enforcement mechanisms (private sector as well as state) make the EU GI system a market-closing mechanism. However, this was contrasted with more competitive elements familiar to the trademark protection process:

- Creation of product specifications
- Registration procedure
- Existence of a public register

It is undisputed that during the introduction process, the member states were oriented towards the so-called *Romanic* concept of origin protection [22,53,100] meaning basically the protection of qualified type of origin-related designations (terroir concept). With Regulation (EEC) No 2081/1992, however, two concepts were established, which can be summarised as a narrow (PDO) and a broad (PGI) terroir concept [63] (pp. 88ff, 127ff).

The narrow, French PDO understanding is based more on the assumption of an objective spatial structural connection, while the broad, German PGI understanding is based more on the recognition of (inter-)subjective traceability (consumer reputation; [97]). The original proposal for a regulation at the end of 1991/beginning of 1992 envisaged one category—namely PDO.

Consequently, many German designations would not have been eligible for protection at the community level [97], [101] (p. 4). The existing protection of German designations abroad would also have been uncertain (ibid.). Therefore, Germany rejected this proposal and insisted on a regulation that was also open to German designations [102]

(p. 194). In this context, it had been important to convince other delegations—especially France—of the German position [102]. Thus, the PGI concept was implemented, which also functioned for the designations recognised under German law and was rather oriented towards competition policy [102].

In addition to the introduction of the PGI concept, Germany's national representatives also managed to successfully introduce their interests in four other areas: procedures regarding recognition and control, the relationship with trademark protection, the relationship with national protection systems, as well as start-up aids [63] (p. 110ff).

However, the PDO concept, or the underlying assumption of a causal link between a certain geographical area and product quality, was not free of criticism either [103] (p. 43), [104].

Together, the PDO and PGI concepts form a Europeanised understanding of terroir, which reflects the view of the EEC member states at the time with their different legal systems ([63], p. 117ff). With the first GI regulation, it was no longer exclusively recognised that a spatial structural link to a certain food quality existed in an objectifiable way (PDO), but it was sufficient that it was recognised intersubjectively (PGI).

This last point was developed in subsequent years. In the ECJ ruling of 10 November 1992 (Case C-3/91, Turrón de Alicante ruling), the PGI concept was strengthened [15] (p. 67). The stronger position of the consumer is also reflected in the fact that in the case of PGIs the *general business practice,* which in the past was based on manufacturer and expert assessments, is continuously being replaced by the consumer's perception (reputation; recognition in the market; for more details see [63] (pp. 147ff, 171f)).

One point that seems to be in need of legal clarification is the question of geographical references in consumer/trade perception. Specifically, it is a question of whether the country of origin or the country of destination is decisive in the claim for injunctive relief (similarly also [105] (p. 590)). For example, the German Federal Supreme Court ruled on 12 December 2019 in the *Culatello di Parma* case that it was not the perception in the country of origin of the product (Italy) that mattered, but the perception of the German consumer as part of *the European consumer* [105] (p. 589). In this case, the (consumer) perception in the country of destination was decisive.

According to experts, however, there is a tendency in borderline cases to be guided by the consumer's view of the country of origin of the product.

### 4.2. Layer 2: Legitimise and Expand

After the many attempts to exert political influence during the introduction process (see also [15] (p. 85)), the adopted regulation needed legal classification and clarification. The ECJ played an important role here. It stabilised the system as a whole and thus strengthened it vis-à-vis trademark law (see [106] (p. 292)). Essentially, the ECJ case law of the last decades has had the following effects [63] (p. 151f):

- Clarification of the importance of the EU GI system as an independently introduced area in EU law
- Clarification and broad interpretation of the introduced rules on violations of protection with regard to references and other misleading practices
- Extension of less defined, possibly generic food names
- Further definition of (quality-influencing) production steps based on existing practice

Elements of competition policy were already established with the introduction of the first GI regulation in 1992, but the goal of safeguarding intellectual property rights was only explicitly addressed for the first time in Regulation (EU) No 1151/2012, Recital 19 [107]. This is not surprising, as the topic of intellectual property as a whole only gained momentum late: the basis for the *Common IPR Strategy* was laid for the first time in 2000 in Article 17(2) of the Charter of Fundamental Rights of the European Union.

The fact that GIs, which were originally conceived as designation rights, are being developed more and more in the direction of a brand or trademark is illustrated, for example,

by the gradual obligatory use of the logo on the product (see also [108] (p. 211)) and, more recently, on advertising material (Regulation (EU) No 2021/2117 Art. 2 [99]).

How agricultural, trade or competition policies are affected can also be seen from the EU institutions involved in the narrower sense: the Directorate General for the Internal Market, Industry, Entrepreneurship and SMEs (DG GROW) is responsible for the enforcement of intellectual property rights. Agricultural designations are an exception due to their history, so DG AGRI has exclusive competence for GIs at the EU level. In the area of enforcement, however, the two DGs have recently been cooperating more intensively, especially in the development of the *GIview* database, resulting in the integration of GIs into the EUIPO trademark register in a separate section [63] (p. 174ff). This illustrates the influence of trademark law and competition policy.

While experts nowadays repeatedly emphasise the benefits of the relatively strong, trademark-like enforcement mechanism, some also stress the importance of the special role of GIs in the IPR world. They describe GI protection as a "Rural Intellectual Property Right", "protection of farmers' and producers' IPR" and "a specific collective right assigned to a region". Overall, it can be stated that from today's perspective, an agricultural policy framework for the protection of certain origin-related designations was created by the nation-states in the 1990s and established with strong competition policy enforcement mechanisms and this was expanded and manifested by the EC in the following decades.

Already during the introduction process, consumer interests were argued for. This was mainly about satisfying the demand (for regional or traditional products) and providing guidance (Regulation (EEC) No 2081/1992, recitals [98]). Overall, the demand argument is seen in a more differentiated way today:

Regulation (EU) No 1151/2012 mentions further socio-political aspects such as the preservation of cultural heritage (Recital 1) as well as resource and animal protection (Recital 23). With the last consolidation of the Food Regulation (amending Regulation (EU) No 2021/2117), the topic of "sustainability, including the economic, social, environmental and climate sustainability" (Recital 1) was introduced due to the overarching CAP objectives [99]. Consequently, products can now also be protected on the basis of their potential contribution to sustainable development (Regulation (EU) No 1151/2012 Art. 1(2)(b)) [107].

Consumer protection arguments are also used at the enforcement level. For example, GIs as state-certified and controlled products at the production level are also part of the *farm2fork strategy* [109] (p. 12). The *farm2fork strategy* originating from the Directorate General for Health and Food Safety (DG SANTE) aims to make food systems fair, healthy and environmentally friendly [109]. In this context, information on origin is also given importance in the sense of consumer clarity with respect to market control [109] (p. 13). Ensuring traceability and avoiding food fraud can also be seen as "typical value chain thinking" (expert interview). The relevant legal foundations are the Food Information Regulation (Regulation (EU) No 1169/2011 [54]) and the Official Controls Regulation (Regulation (EU) No 2017/625 [110]).

For a long time, food control was primarily concerned with the prevention of health risks, but in recent years consumer deception regarding origin has been seen much more strongly as a separate offence of food crime (*food fraud*; expert interview, [111]). For this reason, national reference centres have been set up within the EU and annual control audits have been carried out within the framework of consumer protection. Furthermore, new European fields of action include the introduction of the so-called *GI ID Cards* to improve market controls and the establishment of the cross-product area database *GIview* [63] (p. 174ff).

Furthermore, the EC promotes advertising campaigns for GI products within the framework of EU promotion policy under Regulation (EU) No 1144/2014 [112]. EU promotion policy traditionally has an export focus, but also raises awareness of the system within the EU. It does not focus exclusively, but specifically on GIs (expert interviews). In 2005, more than 30% of the EU promotion policy budget went to food GIs (excluding wine and spirits; [113] (p. 189)). Overall, the funds for EU promotion policy have been greatly

expanded in the last 15 years or so, although they have been declining over the last three years (from EUR 27.6 million in 2006 to EUR 185.9 million in 2023) [109,114].

Highlighting the so-called *Union message* is seen as the central element of EU co-financed campaigns (FTB, participant observation). This is also where the obligatory logo Enjoy! It's from Europe! (Figure 2) was introduced.

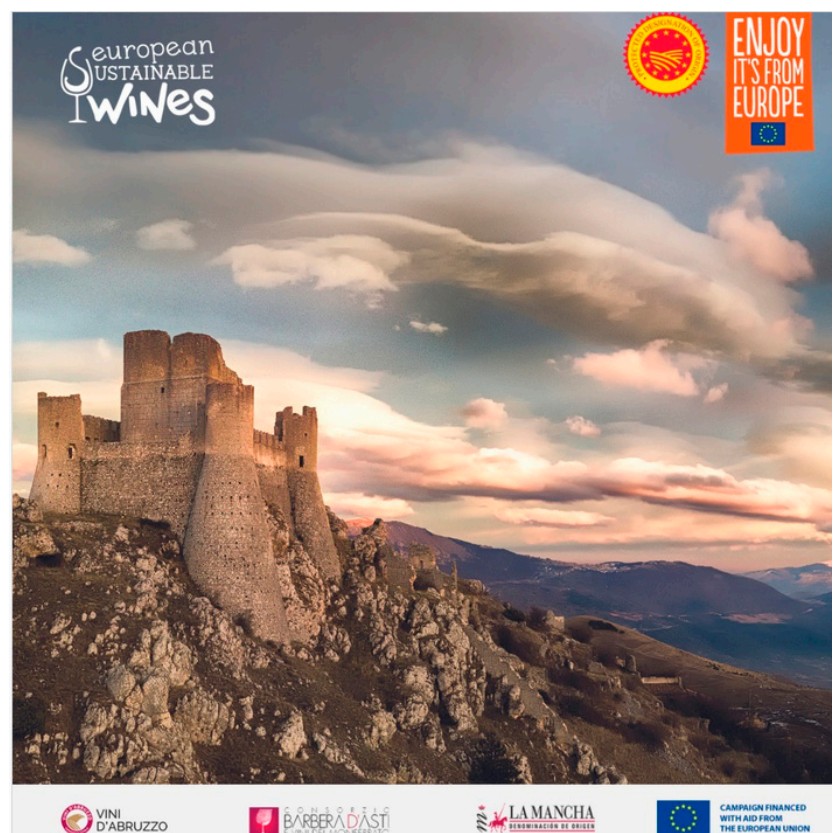

**Figure 2.** Example of an EU co-financed promotion policy programme on Instagram using the logo Enjoy! It's from Europe! Source: Instagram, European Sustainable Wines.

The compulsory use of the EU logo for co-financed promotion activities by the producer associations, also within the EU, illustrates an increasing role of the sender or spokesperson of the EU for regional culinary matters.

Furthermore, the present analysis of GI rules concerning the scope of application shows an expansion in four dimensions (see also [15] (p. 94ff):

(a)    Extension of CAP competence:

The object of Regulation (EU) No 1151/2012 refers—as a CAP policy instrument—in principle only to agricultural products (so-called *Annex I products* (TFEU)) [107]. However, Heine already noted that Regulation (EEC) No 2081/1992 also covered foodstuffs, which the EEC Treaty did not define as an agricultural policy competence [115] (p. 96f). The decisive factor were the economic significance and the inclusion of special cases of the Member States [104,115,116].

(b)    Extension to include product areas without existing GI rules of their own:

The product groups that can be found in the annex to the GI regulation have been continuously expanded. The comparative analysis of the reformed regulations shows that only one product group was deleted (natural mineral and spring water) and 16 product groups (e.g., pasta, salt and chocolate) were included (own analysis).

(c)    Addition of product groups with existing, own GI rules:

Since around the mid-2000s, harmonisation processes have been increasing in regulatory areas of products that traditionally have their own rules of origin protection (aromatised wine products, wine, spirits). In a way, this can be seen as a broadening of the scope. When the experts talk about *harmonisation of the GI regulations*, they are mostly talking about procedural alignment with Regulation (EU) No 1151/2012, in particular concerning product specifications, the registration procedure and the public register (see Supplementary). While there is a general tendency towards alignment with the Food Regulation (EU) No 1151/2012, in some cases sectoral regulations are also taken into account: For example, the reform proposals of the EC with regard to the topic of "goods in transit" are oriented towards regulations for spirits (Regulation (EU) No 2019/787 [117]). Harmonisation processes became very clear in the recent complete integration of aromatised wine products into Regulation (EU) No 1151/2012. DG AGRI plans to further harmonise and simplify the GI system across sectors (expert interviews). Since 2019, a uniform GI regulation has also been under discussion (expert interviews). In March 2022, the Commission presented a first draft of a regulation that merges the regulations previously spread across several legal bases [118]. This has since been updated in May [119].

(d)    Non-restriction through vague definitions and the avoidance of definitions:

Sometimes the avoidance of definitions or the preservation of interpretative leeway is also conducive to the expansion of competence. For example, this is the case when dealing with so-called generic terms that cannot be protected [22,51,115,120] and deletions, both of which potentially devalue a term forever (expert interviews, [15]).

Taking a closer look at the harmonisation processes, the cross-sectoral (agricultural products and food, wines and spirits) comparative analysis of the GI rules showed that spatially relevant categories were hardly harmonised (see Table 3).

Table 3 shows that the protection concept is in principle based on the functional argument of typicality in all three product areas. However, while for food and agricultural products as well as wine very similar PDO and PGI concepts have been introduced, for spirits the more loosely justified link based on a broader GI concept (TRIPS) applies. However, the fact that the broader spirit GI concept and the PGI concept are to be interpreted in a relatively similar way is indicated by the possibility of labelling protected spirits with the PGI logo as well.

When considering the possible territorial scope, it becomes clear that in the case of foodstuffs and wines, this can only extend to entire states in exceptional cases—and in fact, this almost never happens. In the case of spirits, the reference to states is not only traditionally possible, but also common practice. However, the typicality of the product, which refers, for example, to the special soil and climate conditions, is more difficult to prove in the case of larger areas such as nation-states with potentially heterogeneous characteristics (expert interviews).

The comparative analysis with regard to regional raw material binding shows that different rules exist. A relevant point seems to be, in the case of PGIs a raw material linkage must be justified by its specificity (expert interviews, [119]).

With regard to the geographical definition of production stages or steps of further processing, there are also different formulations in the three regulations. The geographical limitation of the value chain must be factually justified with specificity. This becomes clear at the product or regional level in the application or later court proceedings. For example, the Schutzverband der Nürnberger Bratwurst e. V. (Nuremberg Bratwurst Protection Association) successfully argued that packaging at the place of production contributes to maintaining quality, while in the wine sector, it is discussed under which conditions bottling is a quality-determining aspect (expert interviews).

**Table 3.** Spatial references in three * basic GI regulations.

| | Agricultural Products and Foodstuffs | Wine | Spirits |
|---|---|---|---|
| Regulation | (EU) No 1151/2012 | (EU) No 1308/2013 | (EU) No 2019/787 |
| Protection schemes | PDO, PGI | PDO, PGI | GI |
| Justification of the link to quality and origin | PDO: quality or characteristics "essentially" or "exclusively" attributable to "geographical environment" PGI: quality, reputation or other characteristic is "essentially attributable to its geographical origin" (Art. 5) | PDO: quality or characteristics "essentially" or "exclusively" attributable to "geographical environment" PGI: specific quality, reputation or other characteristics attributable to that geographical origin (Art. 93) | "quality, reputation or other characteristic of that spirit drink is essentially attributable to its geographical origin" (Art. 3) |
| Territorial Scope | PDO: place, region, exception: country PGI: place, region, country (Art. 5) | PDO/PGI: region, specific place, exception: country (Art. 93) | Locality, region or country (Art. 3) |
| Regional raw material binding | PDO: 100 % ** PGI.: not explicitly regulated *** | PDO: 100 % PGI: 85 % of grapes (Art. 93) | not explicitly regulated |
| Regional processing | PDO: all production steps PGI: "at least one of the production steps" (Art. 5) | PDO: "all the operations involved, from the harvesting of the grapes to the completion of the wine-making processes" PGI: "its production takes place in that geographical area" (Art. 93) | "the production steps which give the spirit drink the quality, reputation or other characteristic that is essentially attributable to its geographical origin, take place in the relevant geographical area" (Art. 35) |

* Due to the integration of the regulation for aromatised wine products and its relatively small scope of application, this was excluded from the analysis [107,117,121]. ** For animal products incl. feed (if technically not possible: up to 50 % from outside; Regulation (EU) No 664/2014 [122]). *** Can result from the justification of typicality (reputation). Source: own representation

*4.3. Layer 3: Open and Defend*

Overall, the reforms of the regulations also show the increasing influence of trade policy. Regulation (EEC) No 2081/1992 states in the recitals that "provision should be made for trade with third countries" [98], but HEINE (1993) already stated that third countries had to virtually adopt the community system [115] (p. 103). As a consequence, there was a WTO dispute settlement procedure, which led to a greater opening of the system through Regulation (EC) No 510/2006 (USA/Canada v. EU; expert interviews), [120,123]. This second reform thus opened the third phase, which is characterised by system opening and defence.

At the system level, GI law is opposed to trademark law. The American GI critic K. William WATSON even describes these processes as a "battle" that takes place on different "fronts" [124] (p. 8). On the one hand, the parties are concerned with defending their own territory, but on the other hand, they are also concerned with *exporting* their own protection concept. However, this is not only about the directly negotiating partners, but rather about displacing other parties in certain markets or securing export markets (so also [124]).

The overall trade policy influence on the EU GI system can be clearly seen through its international expansion and accompanying opening and defence processes, which are described in more detail in the following paragraphs. As summarised by an EC representative at an event, the internationalisation of GIs has three objectives: the defence of intellectual property, the idea of creating value locally as well as protecting it globally and helping producers to better reach international markets (participant observation).

Overall, the EC's spokesperson role vis-à-vis third countries is particularly evident in five fields of action: FTAs and special GI agreements, multilateral agreements, development cooperation, system enforcement and export promotion.

The EU has 34 agreements concluded or in force with 47 states, from which 1588 designations have been protected within the EU, and 16 ongoing negotiations (as of 13 May 2019; [125], participant observation). In these negotiations, EC representatives act as spokespersons for EU GIs. The aim is to "bring GI law to the world" (expert interview).

EC members repeatedly emphasise the sensitivity of the issue in free trade agreements, describing it, for example, as a "deal breaker", a trigger for "dramas"—although GIs generally only make up a "very small part in very comprehensive free trade agreements". Because of this sensitivity, negotiations are often pushed into the final phase, where several points of contention end up. Thus, GIs sometimes become "bargaining chips" that are used to achieve other goals, "completely different, unrelated topics" such as export quotas. In order not to fall victim to such packaged solutions, agreements that deal with the issue separately, as in the case of China, would be preferable. This would make the issue workable, GI experts would negotiate and the larger free trade agreement would be relieved (expert interviews).

During recent years, the EU has gone through a learning process in negotiations (expert interview, FTB). The strategy of the EC staff seems to be successful: due to the free trade agreements with Canada, Japan and Singapore, a GI system has been established there (expert interview, participant observation).

When protecting designations in the context of negotiations with third countries, the question of orientation towards the place of origin or destination—the geographical reference—also plays a role. In addition to the product's reputation, which is particularly relevant in the European market, other aspects (e.g., language and writing system) can be decisive for the value of the protection (e.g., Japanese 神戸牛 for *Kobe beef*). Normally, in negotiations with third countries, the original names are protected in the national language. In the case of CETA, the protection communities of *Prosecco* and *Bavarian beer* have therefore opted for trademark protection (expert interviews).

The formerly most important agreement for GIs at the multilateral level, the so-called TRIPS Agreement (WTO has been stalled since 1995).

These days the Lisbon Agreement for the Protection of Appellations of Origin and their International Registration is seen as the more relevant multilateral agreement for European interests (expert interviews).

The negotiation process started in 2008 (expert interview) and was supplemented by the Geneva Act, which entered into force on 26 February 2020. It required five members to do so (in comparison TRIPS listed one hundred and sixty-four signatories as of 7 February 2020, [126], expert interviews). The EU's objectives during the negotiations were largely achieved and referred to

- The modernisation of the legal framework
- The admission of supranational confederations of states
- The extension of the scope of application to include PGIs (expert interviews)

On further consideration, by opening up or expanding the system, not only can producers of traditional foods from Europe better reach international markets, but also producers from regions in non-EU countries can better place their products on the European market. The latter was, after all, the impetus for the second reform of GI law in the sense of opening up the system. However, third-country products recognised by the EU within the framework of free trade or multilateral agreements cannot bear the EU logos (expert interview).

GI protection can certainly open up new market opportunities and thus have a positive impact on regional development, as the example of *Café de Valdesia PDO* shows [127]. Here, registration as a PDO contributed to strategic quality work, more intensive cooperation along the value chain, better visibility and image enhancement of the product, new investments in local structures and sustainable regional development [127].

With regard to development policy aspects, there is also GI cooperation with, for example, the African Union. Currently, on the initiative of the African Union, potentially protectable products are being identified together with local actors, the FAO and the EU-IPO, and the basis for registration is being created (expert interview). Attempts are also being made to call on funding from the EU's development policy (expert interview).

A total of 217 products from third countries are currently listed in eAmbrosia (as of 8 November 2022), i.e., in addition to the designations recognised in political agreements. Moreover, the fact that 47 new applications are also in progress shows not only that direct registration can be interesting in the context of development policy, but that the full adoption of the EU regulatory system is useful for many an international producer group.

Furthermore, enforcement or control mechanisms concerning GIs have been strengthened at the EU level with competition and consumer protection policy instruments, as already mentioned. However, an interest in trade policy also reveals new European fields of action in the area of control. Thus, cross-border issues affecting trade with third countries, such as *goods in transit* and *abuse of names on the internet*, became more important. They have since been incorporated into the revised Regulation (EU) No 1151/2012 and illustrate the increasing institutionalisation of control-related fields of action at the EU level and thus the strengthening of protection.

## 5. Discussion

The basis of the EU's spokesperson role for GI was gradually laid through old protection rights and manifested with the introduction of the first GI regulation in 1992. Since the introduction of systematised GI rules at the EU level some 30 years ago, the system has become more institutionalised in all allocation dimensions considered (competences, control, resources, interaction and legitimacy). The role of the EC as a spokesperson for regional culinary matters has been expanded in and outside the EU, e.g., in free trade, consumer protection and competition policy. DG AGRI is the key actor because it has exclusive competence at the European level. Enabling conditions (path dependencies) and common interests across policy fields (discourse coalitions) were crucial for this development.

With regard to European integration, understood as an interplay of territorial and relational processes [6], it can be stated that it is discussed to which extent national or European competence exists for the regulation of origin-related designations in a broader sense, which means that there is probably "pooled-territoriality" [6] (p. 2705).

The results illustrate that the complex processes at the EU level are characterised above all by sectoral as well as multiscale influences. Moreover, although the EU GI system has never lost its agricultural policy focus, it has been increasingly influenced by other policy fields in recent decades and has developed above all in the direction of *IPR*.

The increase in the European level's scope for action is clearly evident in the area of enforcement and control. Here, on the one hand, competition policy institutions played a more important role; on the other hand, consumer policy instruments have also been expanded. They are intended to improve the strength of enforcement at the regional level, on the one hand by strengthening GI producer groups as rights holders, and on the other hand by strengthening (nationally) state-coordinated market control.

These central points raise in particular the questions of what role "spillover effects" (first [128]) played in the integration of the EU GI system, to what extent spatial references were used discursively and what potentials arise from strengthening the regional level or the functionally delimited speciality regions (soft spaces).

In its role as a regulator for consumer protection-related market controls, the EC—or more specifically DG SANTE—has also increased the enforcement power of the system in the intra-European market. DG AGRI has also increased enforcement in international trade (e.g., on the internet). These actions concerning controls arise from the discourse coalition of agricultural, consumer protection and competition policy interests and the

recognition of EU GIs as *Rural IPR*. This seems to be a spillover effect, which basically describes the institutional radiating from one limited policy field into other fields.

The strengthening and harmonisation of the system within Europe since the mid-2000s can also be seen in connection with global integration processes and its assertion against trademark law. It is not clear to what extent competitive logics are due to impulses from global or intra-European trade. It seems logical, however, that the EC is better able to enter into international negotiations and conduct advertising campaigns with a unified, coherent system than with a sectorally fragmented one. These processes can also be seen as an expression of the attempt to speak with one voice. At the same time, these harmonisation processes are accompanied by the loss of special roles (in more detail [63] (p. 165f)). The moving context could affect the attractiveness of GIs and hence rural development [129].

The tendency towards "supra-territoriality" [6] (p. 2705) can also be seen in the increasing spokesperson activities of the EC for GIs in relation to the restrictions under EU state aid law for simple and combined origin-related designations. On the other hand, the role of the EC as a spokesperson for origin-specific culinary affairs also depends on how the possibilities of *national* origin labelling change. These are currently the subject of intense discussion. For example, Germany, Austria and Sweden are currently discussing mandatory origin labelling in certain areas (e.g., food retail, out-of-home catering) or for certain products (e.g., pork), while in France and Italy, this has already been implemented and expanded.

The comparative analysis of directly spatially relevant categories across the sectors of *agricultural products and food*, *spirits* and *wine* has shown that these have not yet been harmonised or, in other words, discursively excluded ("discursive deterritorialization" [8]). The harmonisation largely concerns the procedures. Spatial references (especially raw material binding and local consumer expectations) usually come back into the discussion at a later point in time, namely at the regional level or in concrete disputes ("discursive reterritorialisation" [8]). The classification in the quality programmes with Regulation (EU) No 1151/2012 can also be interpreted as a deterritorialisation process because the aspect of origin is pushed into the background.

In the political process, spatial references in the GI system are often still used today as a category of distinction (*Europe of North and South*), and in the national context of Germany, mainly as an argument for defending the system (*Romanic System*; *System of the Others*; for more details: [63] (p. 241ff)). However, Germany has already helped to shape the system under the mentioned five important points during the introduction process, including the PGI concept.

Another point where spatial references could play a decisive and also economically relevant role is that of consumer perception. However, it has not yet been legally clarified whether the perception of consumers in the country of origin or in the country of protection is decisive for protection in the case of a lawsuit [63,105].

Furthermore, speciality regions, meaning regional territorial demarcations for certain PDO/PGI/GI products, can also be operationalised as soft spaces with their European framework and their functional founding logic. The functional link between certain spatial conditions at a certain point in time and a specific product quality is the core of the terroir principle on which the idea of GI protection is based.

Firstly, this link is expressed in the consumers' spatial appreciation, which is the basis for protection, and which is shown in the market through the act of purchase or a willingness to pay more. Secondly, there are the functional arguments that have to be put forward by the producer groups in the context of the protection application and which relate, for example, to soil and climate conditions and to the space-specific reputation of the product among consumers.

What distinguishes speciality regions from other soft spaces such as macro-regions is that they are currently unlimited in time. They are also less informal spatial concepts [130] (p. 83), but rather strongly formalised through their legal definition.

However, like other soft spaces, speciality regions can be seen as "testing grounds" [6] (p. 4) and new cooperation spaces. The GI system is relevant as a sustainability innovation in that it can reduce the risk of the (global) market's focus on quantity and price in order to protect traditional foods. Spatial characteristics that are justified by traditional—or more recently also explicitly sustainable—production methods (e.g., preservation of diversity, environmental protection, animal welfare) are relevant to specification.

With the strengthening of the regional level (especially producer groups as rights holders, but also regional enforcement of market control in the sense of consumer protection), regional actors are also given more responsibility. At the level of the producer groups, there is still a need for clarification regarding the organisational forms and the necessity of recognition [63] (p. 172f), [131,131].

The example of Germany, however, clearly shows that where other systems are already established—which, unlike the GI system, are based, for example, on state-established quality criteria and administrative boundaries—the enforcement of the GI policy is more difficult than in other member states [63] (p. 258ff).

The support of governments and administrations can play a key role in the legitimisation process of the EU GI system in national and regional contexts [46,63,132,133], but there is also evidence that the timing of EU accession influences the chances of protection [119] and [63] (p. 157f). GIs can be an instrument for regional agricultural policy to promote a quality strategy, alongside national quality programmes and the promotion of regional collective and certification marks [63,134].

## 6. Conclusions

Looking at the limitations of the study, it can be stated that since the study refers to European integration processes and therefore focused on intra-European experts and sources, no conclusive statements can be made about the influence of international or multilateral institutions.

Furthermore, a more political science perspective on the described spillover effect could provide further insights for the integration debate.

Moreover, in order to explore the actual connectivity to structures (e.g., legal frameworks, associations) and interests in implementation processes, a comparison of regional practices based on themes would be valuable in the context of border studies, where the national context has recently been seen more as an explanatory factor for governance structures [135]. The need for further development of GI-specific measurement methods with the help of greater consideration of the respective context has also been stated from a more legal perspective [132] (p. 56ff). The "simplified notion of reputation" [132] (p. 59) oriented towards trademark law, which is based on sales figures or consumer surveys at a certain point in time, does not do justice to the required continuity or the "ongoing vitality" [132] (p. 56) of cultural developments.

Particularly demanding, therefore, but also relevant, seems to be the institutional integration of sustainability goals, which contains a clearly future-oriented time component. Further scale-sensitive studies that understand speciality regions even more strongly as cooperation areas could provide better insights at this point.

It remains to be seen whether the EC, in designing the EU GI system, as suggested by Regulation (EU) No 1151/2012, consolidated on 8 July 2022, and the new EC proposal for a unified GI regulation of 2 May 2022, will be more strongly oriented towards competition policy mechanisms and administered, for example, as pure IP law at EUIPO, or whether it will be further developed as an independent system. The latter would have the potential to give greater weight to cultural embedding and other aspects relevant to sustainability and specific to the region, such as those currently demanded by society, and not to reduce them to *purely legal issues*.

At the regional level, the EU GI system is a benefit for consumers, producers and regional development. It offers clearly defined, publicly accessible, officially controlled and clearly communicated origin criteria—in a uniform system across Europe. Too much

focus on trademark mechanisms seems to contribute little to the *sui generis* legal field of EU GI, the preservation of cultural culinary arts and the further development towards sustainability. In any case, if the goal of protecting regional specificities is to be carried into the future, harmonisation *automatisms* should always be the object of critical reflection.

**Supplementary Materials:** The following supporting information can be downloaded at: https://www.mdpi.com/article/10.3390/su15032666/s1, Table S1: Milestones in the development of GI rules for agricultural products and foodstuffs.

**Author Contributions:** Conceptualisation, investigation, methodology, writing—original draft preparation, project administration, K.S.; writing—review and editing, H.L., R.B. and T.C.; supervision, T.C. and R.B. All authors have read and agreed to the published version of the manuscript.

**Funding:** This research was funded by German Academic Scholarship Foundation (Studienstiftung des deutschen Volkes e. V.) We also acknowledge financial support by Deutsche Forschungsgemeinschaft and Friedrich-Alexander-Universität Erlangen-Nürnberg within the funding programme "Open Access Publication Funding".

**Informed Consent Statement:** Informed consent was obtained from all subjects involved in the study.

**Data Availability Statement:** Publicly available datasets were analyzed in this study. This data can be found here: https://ec.europa.eu/info/food-farming-fisheries/food-safety-and-quality/certification/quality-labels/geographical-indications-register/ (accessed on 28 December 2022).

**Acknowledgments:** We thank all the interviewees who contributed their time and knowledge to this study.

**Conflicts of Interest:** The authors declare no conflict of interest. The funders had no role in the design of the study; in the collection, analyses, or interpretation of data; in the writing of the manuscript; or in the decision to publish the results.

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
