# Peer review of "European Integration Processes in the EU GI System—A Long-Term Review of EU Regulation for GIs"

_sustainability, doi:10.3390/su15032666_

Round 1
Reviewer 1 Report
The introduction is too long and does not state what the paper is about: what is the authors’ research question? what is their hypothesis? We have some clues l.291 but it is a little bit late and not developed enough. Introduction could be shorter and more efficient (context, question, originality of approach, structure of the article) and what is dedicated to GI’s history could be moved to the state of the art. + I don’t really understand what is the role of this long development on quality linked to traceablity as it is not developed futher in the article, and in the results. Maybe the paper should focus its demonstration on GIs.
To my point of view, the German perspective is quite interesting and original, as a non Mediterranean state and so, with no historical protection system like the AOC or DOC. Maybe this, and german history about food quality could be quickly developed in the article?
l.60. if I recall correctly, those national protection systems were also strongly debated within international arenas such as the WTO. See notably Barham, 2003, translating terroir: the global challenge of French AOC labeling, Journal of rural studies. Given the authors’ approach, I think that this part of the GI’s history (and the TRIPs agreement) should be developed as it might take a large part in how the EU appropriate the GI system and made a strong policy of theirs (which you confirm afterwards in your results and discussion).
l.276 and following. We could expect other references such as Bowen’s or maybe publications of Allaire, Sainte-Marie, Vandecandeloere or Marie-Vivien.
Methodology is well presented. l.323: please specify what type of events and expert discussions. L.325: what was the content of those interviews? What main topics were dealt with?
l.598 : “the comparative comparison”
The results are a bit long and it is hard to follow the authors’ demonstration. Maybe some details could be suppressed to make the text easier to read. If you better present your demonstration at the beginning, it might help you keep what is really useful for your argument.
The discussion is well written and propose a global view of the paper, which is welcome. A reference that might be interesting : https://www.sciencedirect.com/science/article/abs/pii/S0305750X15000029
Reviewer 2 Report
The work offers a 360-degree overview of European policies on geographical indications over the last 30 years. From this point of view it represents a very useful work for anyone approaching the subject.
The aspects of the paper that in my opinion need to be improved are the following:
1. although it is a 'review' work, it is still useful to better clarify what the objectives of the paper are, an aspect which then leads to the second -important- observation;
2. the criteria with which the articles cited in the paper were identified in the immense production on the subject are not clear;
3. the limitations of the work are not specified;
4. the authors preferred to propose the 'Overlook' paragraph rather than the more traditional 'Conclusions'. From my point of view the 'conclusions' are always preferable, as they are the place where it includes both the Overlook, but also other key aspects, such as the limitations;
5. it could be appropriate to introduce the theme of food and wine tourism in the Overlook, just as the aspect of sustainability could be better explored
